# Causal relationship between COVID-19 and chronic pain: A mendelian randomization study

**Yuchao Fan** [1‡]*, **Xiao Liang** [2‡]

**1** Department of Anesthesiology, Sichuan Clinical Research Center for Cancer, Sichuan Cancer Hospital & Institute, Sichuan Cancer Center, Affiliated Cancer Hospital of University of Electronic Science and Technology of China, Chengdu, China, **2** Department of Anesthesiology, West China Hospital, Sichuan University, Chengdu, Sichuan Province, China

‡ YF and XL are joint co-first authors on this work.
* yuchaofan_pain@126.com

## Abstract

### Objective

COVID-19 is a highly transmissible disease that can result in long-term symptoms, including chronic pain. However, the mechanisms behind the persistence of long-COVID pain are not yet fully elucidated, highlighting the need for further research to establish causality. Mendelian randomization (MR), a statistical technique for determining a causal relationship between exposure and outcome, has been employed in this study to investigate the association between COVID-19 and chronic pain.

### Material and methods

The IVW, MR Egger, and weighted median methods were employed. Heterogeneity was evaluated using Cochran's Q statistic. MR Egger intercept and MR-PRESSO tests were performed to detect pleiotropy. The Bonferroni method was employed for the correction of multiple testing. R software was used for all statistical analyses.

### Result

Based on the IVW method, hospitalized COVID-19 patients exhibit a higher risk of experiencing lower leg joint pain compared to the normal population. Meanwhile, the associations between COVID-19 hospitalization and back pain, headache, and pain all over the body were suggestive. Additionally, COVID-19 patients requiring hospitalization were found to have a suggestive higher risk of experiencing neck or shoulder pain and pain all over the body compared to those who did not require hospitalization. Patients with severe respiratory-confirmed COVID-19 showed a suggestive increased risk of experiencing pain all over the body compared to the normal population.

**Data Availability Statement:** The basic data for the results of this study can be obtained from the summary data of GWAS (https://gwas.mrcieu.ac. uk/). All data can be accessed without the need for registration or login. Detailed information on all dataset titles is listed in Table 1.

**Funding:** The author(s) received no specific funding for this work.

**Competing interests:** The authors have declared that no competing interests exist.

## Conclusion

Our study highlights the link between COVID-19 severity and pain in different body regions, with implications for targeted interventions to reduce COVID-19 induced chronic pain burden.

## Introduction

COVID-19, a highly infectious respiratory disease caused by the SARS-CoV-2 virus, has emerged as a global pandemic since late 2019, claiming more than 6.5 million confirmed deaths and over 600 million confirmed cases as of current records [1]. Transmission of the virus occurs predominantly via respiratory droplets emitted by infected individuals, as well as through contact with contaminated surfaces, followed by contact with the face or mouth [2]. Symptoms of COVID-19 include fever, cough, fatigue, loss of taste or smell, muscle aches, and shortness of breath, with some patients also experiencing gastrointestinal symptoms, ranging from mild to severe [3, 4]. Older adults individuals and those with underlying health conditions, such as heart disease, diabetes, and obesity, face the highest risk of severe illness and mortality [5–7].

Although most people recover within a few weeks after contracting COVID-19, a subset of individuals experience persistent symptoms and complications that can last for weeks or even months after the initial infection [8, 9]. This condition, known as long- COVID, is a prolonged and distressing health issue that affects a significant proportion of COVID-19 survivors [10]. Pain, in its diverse forms such as headache, muscle pain, joint pain, abdominal pain, and neuropathic pain, is a common manifestation of long- COVID [11]. The type, severity, and duration of pain can greatly vary among individuals, making management and treatment challenging [12, 13]. Although the exact mechanisms of pain in long- COVID remain incompletely understood, they may involve a combination of immune dysregulation, neurosensitivity, and inflammation [14]. Physical and emotional stress, sleep disturbances and reduced physical function may also exacerbate the onset of pain [15, 16].

The debilitating pain experienced by individuals with long-COVID has a profound impact on their quality of life, often leading to depression, anxiety, and other mental health problems [1, 9, 17]. This burden extends beyond the individual and imposes significant costs on society, including increased healthcare expenses, decreased productivity, and diminished quality of life for both patients and their families [18, 19]. Managing the pain associated with long-COVID is crucial to mitigate the physical and emotional toll it exacts. Therefore, focused monitoring and specialized management are essential to prevent and treat pain resulting from COVID-19, ultimately reducing the incidence of long-COVID.

The majority of studies on pain development following COVID-19 infection are observational, with a dearth of high-quality research to identify the specific COVID-19 patient cohorts at greater risk of long-term pain. As such, a comprehensive understanding of the factors that contribute to pain after COVID-19 is yet to be established. While various theories exist to explain the development of persistent pain after COVID-19, more rigorous investigations are necessary to establish a causal relationship between COVID-19 and long-term pain.

Mendelian randomization (MR) is a statistical method employed in epidemiology and genetics to establish causal relationships between exposures and outcomes. This technique employs genetic variants linked to the exposure of interest as IVs to estimate the causal impact of exposure on outcome [20]. MR studies offer various benefits over observational and prospective studies [21–24]. Firstly, they provide robust evidence for causality by mitigating the

influence of confounding factors on genetic variants. Consequently, MR studies can distinguish between potentially causal and non-causal associations. Secondly, MR studies are less prone to reverse causality bias, commonly encountered in observational studies. Thirdly, MR studies enable researchers to explore potential causal pathways for the association between exposure and outcome, providing insight into the biological mechanisms by which exposure influences outcome and identifying novel targets for intervention. Finally, MR studies can harness data from large-scale genetic consortia, offering high statistical power to detect causal effects. This study employs the MR technique to investigate the causal link between COVID-19 and pain, a potent tool to inform the development of interventions and treatments aimed at reducing the incidence of long-COVID pain.

## Material and methods

### Study design

This study employed a two-sample MR approach utilizing summary statistics from genome-wide association studies (GWAS) to investigate the potential causal association between different conditions with COVID-19 and bodily pain in various regions. The investigation aimed to explore the relationship between different conditions with COVID-19 as the exposure and bodily pain as the outcome, while circumventing the need for ethical approval through the use of publicly accessible data. To satisfy the IVs assumptions in MR, the study utilized genetic variation to estimate the causal effect of the exposure on the outcome. The essential criteria for genetic variation to fulfill the IVs assumptions in this study are as follows Fig 1): (1) The genetic variant must exhibit an association with the exposure; (2) the genetic variant must not exhibit an association with any confounding factors that may influence the association between the exposure and outcome; and (3) the genetic variant should not directly influence the outcome, except through its association with the exposure.

### Data source

This study relied on genetic associations derived from independent GWAS datasets featuring the same ancestral population, in order to account for confounding factors. The COVID-19

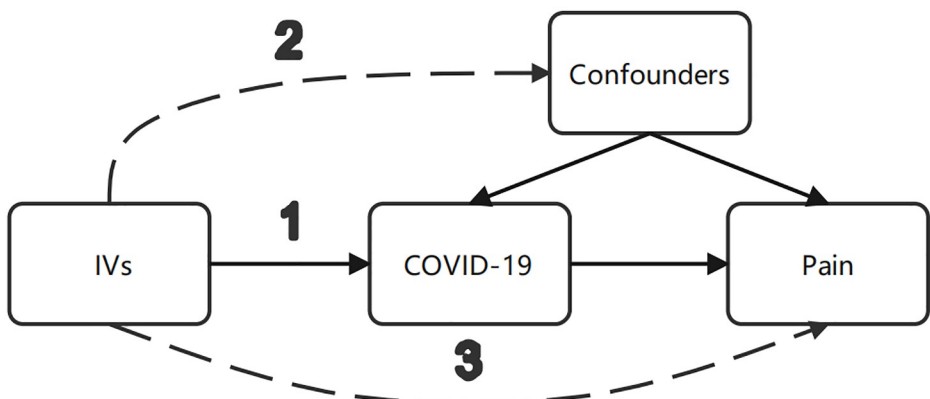

**Fig 1. MR study design examining the causal effect of different conditions with COVID-19 on bodily pain in various regions, requiring a genetic variant that meets three criteria: (1) The genetic variant must exhibit an association with the exposure; (2) the genetic variant must not exhibit an association with any confounding factors that may influence the association between the exposure and outcome; and (3) the genetic variant should not directly influence the outcome, except through its association with the exposure.**

GWAS dataset comprised four separate datasets sourced from European populations, which reflected varying levels of COVID-19 prevalence. These datasets included COVID-19 (infection vs. normal population), COVID-19 (hospitalized vs. normal population), COVID-19 (hospitalized vs. not hospitalized), and COVID-19 (very severe respiratory confirmed vs. normal population). The 13 pain-related datasets used in this study encompassed Pain in joint, Pain in joint (Lower leg), Low back pain, Low back pain (Lumbar region), Pain in limb (Lower leg), Back pain, Facial pain, Headache, Hip pain, Knee pain, Neck or shoulder pain, Stomach or abdominal pain, and Pain all over the body. These datasets reflected the prevalence of pain in the study population. Although pain is experienced throughout the body and some of the datasets depict overlapping pain sites, they complement each other by either identifying distinct pain sites or localizing to more specific areas. For example, while back pain and low back pain may overlap in certain areas, there are also areas of back pain that are not represented in low back pain. On the other hand, Low back pain (Lumbar region) corresponds to a more specific area of low back pain. A detailed overview of the datasets featured in this study is provided in Table 1.

## Selection of instrumental variables

In this investigation, the selection of Instrumental variables (IVs) was conducted meticulously, adhering to strict criteria. Single-nucleotide polymorphisms (SNPs) were deemed valid IVs if they exhibited significant genome-wide association with the exposure ($P < 1e-5$) [25]. Moreover, SNPs with a minor allele frequency (MAF) greater than 0.01 in the outcome and a linkage disequilibrium (LD) r2 of less than 0.001 within a 10,000 kb distance were selected as IVs.

**Table 1. Details of the datasets included in the study.**

| Trait | Variable | GWAS ID | No. | | | | | Population | Category |
|---|---|---|---|---|---|---|---|---|---|
| | | | Year | Cases | Controls | Sample size | SNPs | | |
| Covid-19 | COVID-19 (infection vs. normal population) | ebi-a-GCST011073 | 2020 | 38984 | 1644784 | 1683768 | 8660177 | European | NA |
| | COVID-19 (hospitalized vs. normal population) | ebi-a-GCST011081 | 2020 | 9986 | 1877672 | 1887658 | 8107040 | European | NA |
| | COVID-19 (hospitalized vs. not hospitalized) | ebi-a-GCST011080 | 2020 | 4829 | 11816 | 16645 | 8360206 | European | NA |
| | COVID-19 (very severe respiratory confirmed vs. normal population) | ebi-a-GCST011075 | 2020 | 5101 | 1383241 | 1388342 | 9739225 | European | NA |
| Pain | Pain in joint | ukb-b-13019 | 2018 | 1451 | 461559 | 463010 | 9851867 | European | Binary |
| | Pain in joint (Lower leg) | ukb-b-2083 | 2018 | 1624 | 461386 | 463010 | 9851867 | European | Binary |
| | Low back pain | ukb-b-1557 | 2018 | 2439 | 460571 | 463010 | 9851867 | European | Binary |
| | Low back pain (Lumbar region) | ukb-b-19813 | 2018 | 1423 | 461587 | 463010 | 9851867 | European | Binary |
| | Pain in limb (Lower leg) | ukb-b-9708 | 2018 | 1689 | 461321 | 463010 | 9851867 | European | Binary |
| | Back pain | ukb-b-9838 | 2018 | 118471 | 343386 | 461857 | 9851867 | European | Binary |
| | Facial pain | ukb-b-17107 | 2018 | 8595 | 453262 | 461857 | 9851867 | European | Binary |
| | Headache | ukb-b-12181 | 2018 | 93308 | 368549 | 461857 | 9851867 | European | Binary |
| | Hip pain | ukb-b-7289 | 2018 | 52087 | 409770 | 461857 | 9851867 | European | Binary |
| | Knee pain | ukb-b-16254 | 2018 | 98704 | 363153 | 461857 | 9851867 | European | Binary |
| | Neck or shoulder pain | ukb-b-18596 | 2018 | 106521 | 355336 | 461857 | 9851867 | European | Binary |
| | Stomach or abdominal pain | ukb-b-11413 | 2018 | 39646 | 422211 | 461857 | 9851867 | European | Binary |
| | Pain all over the body | ukb-a-477 | 2017 | 5099 | 331551 | 336650 | 10894596 | European | Binary |

GWAS, genome-wide association studies; NO, number; SNP, Single nucleotide polymorphism

SNPs linked with confounding factors or outcomes according to the Phenoscanner database (http://www.phenoscanner.medschl.cam.ac.uk/) were excluded from the study. To assess the strength of the IVs, we calculated the proportion of variance accounted for by each SNP, and the F-statistic. A value of less than 10 for the F-statistic indicated the selected genetic variants were weak IVs, which could result in biased outcomes [26]. Thus, we cautiously interpreted the findings. The determination of the attributable fraction of variance ascribed to individual SNPs was computed through the formula: $R^2 = 2 \times \beta^2 \times EAF \times (1 - EAF)/(2 \times \beta^2 \times EAF \times (1 - EAF) + 2 \times SE^2 \times N \times EAF \times (1 - EAF))$. Simultaneously, the F-statistic was ascertained using the following equation: $F = (N - k - 1)/k \times R^2/(1 - R^2)$, wherein 'N' denotes the count of samples subjected to the GWAS, 'k' signifies the number of IVs, and 'R²' characterizes the degree to which IVs expound upon the exposure under investigation.

## MR analysis

To explore the potential causal link between different conditions with COVID-19 and pain in distinct anatomical regions, we employed three distinct MR methodologies. The primary approach was the Inverse Variance Weighting (IVW) method, while we employed MR-Egger and weighted median methods as complementary measures. Significant substantive outcomes necessitated concurrence among the results derived from MR-Egger, the weighted median, and IVW methodologies in terms of directional implications. In the absence of such alignment, the observed significance would remain predominantly nominal in nature.

To mitigate the challenge associated with conducting numerous statistical tests, we implemented a stringent correction method based on the Bonferroni principle, setting the significance threshold at $p < 0.0125$ (0.05 divided by 4). P-values falling within the interval of 0.0125 to 0.05 were deemed indicative of preliminary indications of potential causal relationships, warranting subsequent validation and confirmation. The effect estimates were presented as odds ratios (ORs) with corresponding 95% confidence intervals (CIs).

We employed Cochran's Q statistic, using MR-Egger and IVW methods, to evaluate heterogeneity [27, 28]. A p-value greater than 0.05 indicated the absence of heterogeneity. To assess the influence of individual SNPs on the causal relationship between exposure and outcome, we conducted a "leave-one-out" analysis. In situations where heterogeneity existed, we utilized a random effects model to estimate the causal association. Moreover, we performed MR-Egger intercept and MR-PRESSO tests to detect possible pleiotropy of IVs [29]. A p-value greater than 0.05 was regarded as evidence of the absence of pleiotropy. The statistical analyses for all MR procedures were carried out using the R software (version 4.2.2) and R Package "TwoSampleMR" and "MRPRESSO."

## Results

The aim of our study was to examine the causal relationship between various COVID-19 conditions and bodily pain in different regions, using a two-sample MR analysis approach. This method allowed us to investigate whether COVID-19 conditions were a potential causal factor for experiencing pain in different parts of the body. In adherence to our stringent criteria for SNP selection, a defined set of SNPs has been meticulously curated to function as IVs in the context of our MR study. Detailed information pertaining to these chosen SNPs, which includes Number of SNPs extracted from exposure, Number of SNPs extracted from outcome, Number of SNPs after harmonising with outcome, SNPs removed due to harmonization, and Filtering for SNPs significantly associated with exposure in outcome data, has been meticulously documented in S1 Table for comprehensive reference. No SNP were removed due to MAF.

The primary method employed to assess causal effect was the IVW method, which revealed that although COVID-19 patients do not exhibit a correlation with pain risk overall, certain cases of COVID-19 are associated with an elevated risk of pain (S2 Table, Fig 2B). Specifically, we found that patients hospitalized with COVID-19 were at a higher risk of developing various types of pain compared to the normal population. The odds ratios (ORs) for Pain in joint (Lower leg), Back pain, Headache, Knee pain, Neck or shoulder pain, and Pain all over the body were 1.001104 (95% CI: 1.000484–1.001725; IVW P value = 0.000487), 1.003429 (95% CI: 1.000014–1.006855; IVW P value = 0.049055), 1.003588 (95% CI: 1.000320–1.006867; IVW P value = 0.031366), 1.004784 (95% CI: 1.001635–1.007942; IVW P value = 0.002877), 1.002835 (95% CI: 1.000078–1.005599; IVW P value = 0.043864), and 1.001085 (95% CI: 1.000178–1.001994; IVW P value = 0.019048), respectively (Figs 3A–3D, 4A–4D, Table 2). These results indicate that COVID-19 patients who require hospitalization are at greater risk of experiencing pain in different parts of the body than those in the general population. However, owing to the incongruity in the directional outcomes between IVW and MR-Egger analyses, the increased risk of knee pain and neck or shoulder pain among patients hospitalized with COVID-19 compared to the normal population was merely nominal (Table 2). Furthermore, considering the Bonferroni-corrected threshold set at 0.0125 for multiple testing, it is noteworthy that patients hospitalized with COVID-19 exhibit a significantly increased risk of

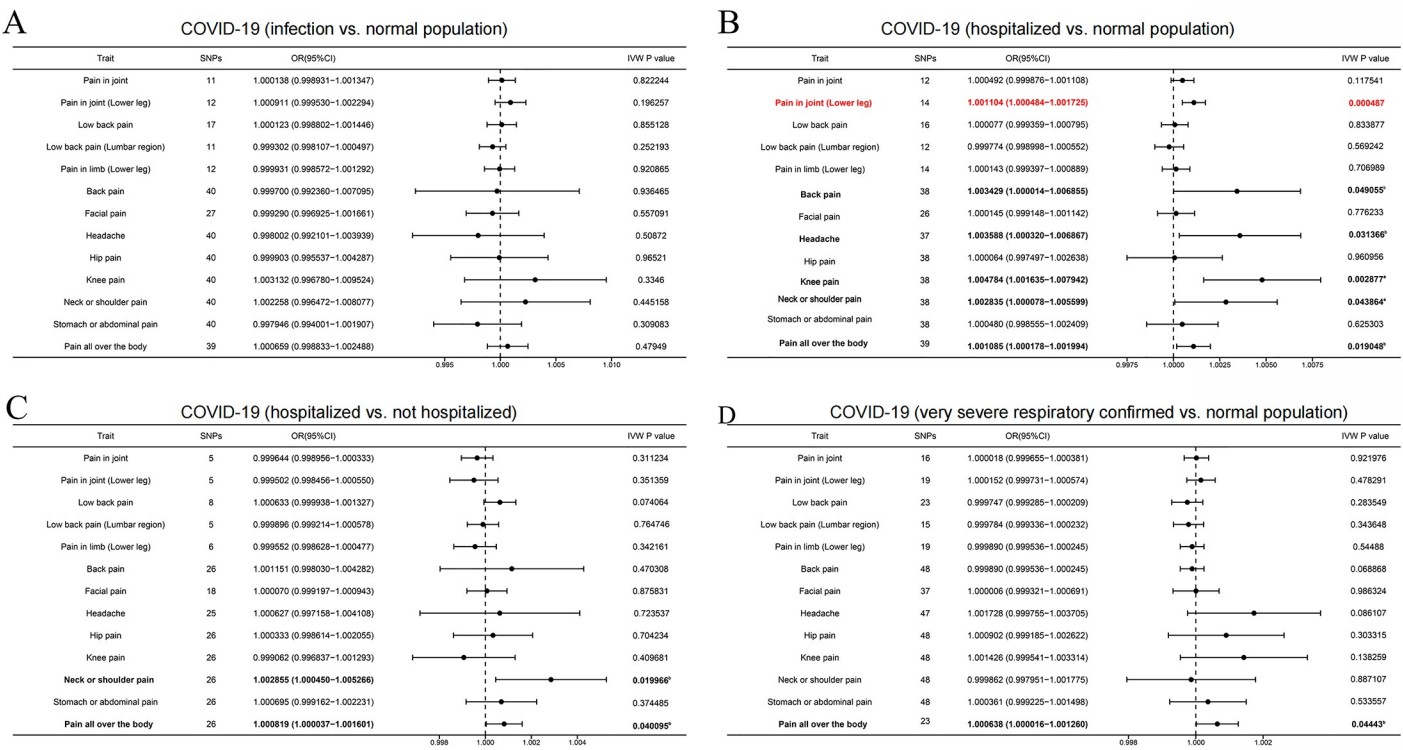

**Fig 2. Forest plot of the potential causal association between different conditions with COVID-19 and bodily pain in various regions. (A)** COVID-19 (infection vs. normal population), **(B)** COVID-19 (hospitalized vs. normal population), **(C)** COVID-19 (hospitalized vs. not hospitalized), and **(D)** COVID-19 (very severe respiratory confirmed vs. normal population). The different conditions with COVID-19 has been identified as a potential risk factor in the pain occurrence, as indicated by its highlighted representation in Red and Bolded font. The symbol 'a' signifies that in the context of MR-inverse-variance weighted methods, the direction of results deviates from the outcomes observed in other methodologies, indicating a mere nominal association. The symbol 'b' signifies a inverse-variance weighted p-value less than 0.05 but greater than 0.0125, indicating that there is a suggestive association between the condition with COVID-19 and pain.

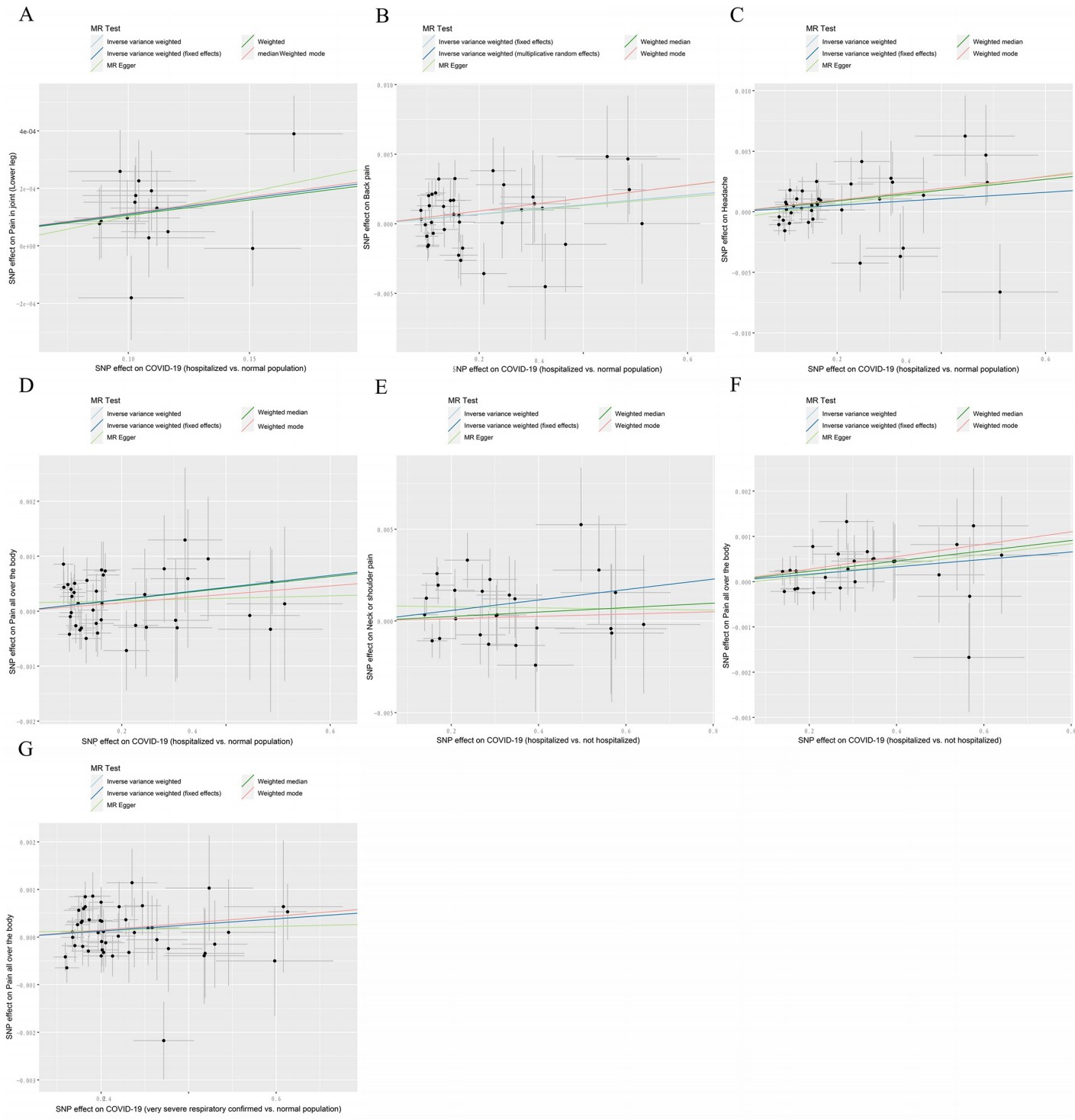

**Fig 3. Scatter plot displaying individual MR estimates of the effect of SNPs on COVID-19 and pain risk, including regression lines for different MR methods.**
COVID-19 (hospitalized vs. normal population) on (**A**) Pain in joint(Lower leg), (**B**) Back pain, (**C**) Headache, (**D**) Pain all over the body; COVID-19 (hospitalized vs. not hospitalized) on (**E**) Neck or shoulder pain, (**F**) Pain all over the body; (**G**) COVID-19 (very severe respiratory confirmed vs. normal population) on Pain all over the body. IV, instrumental variant; IVW, inverse variance weighted; MR, Mendelian randomization; SNP, single-nucleotide polymorphism.

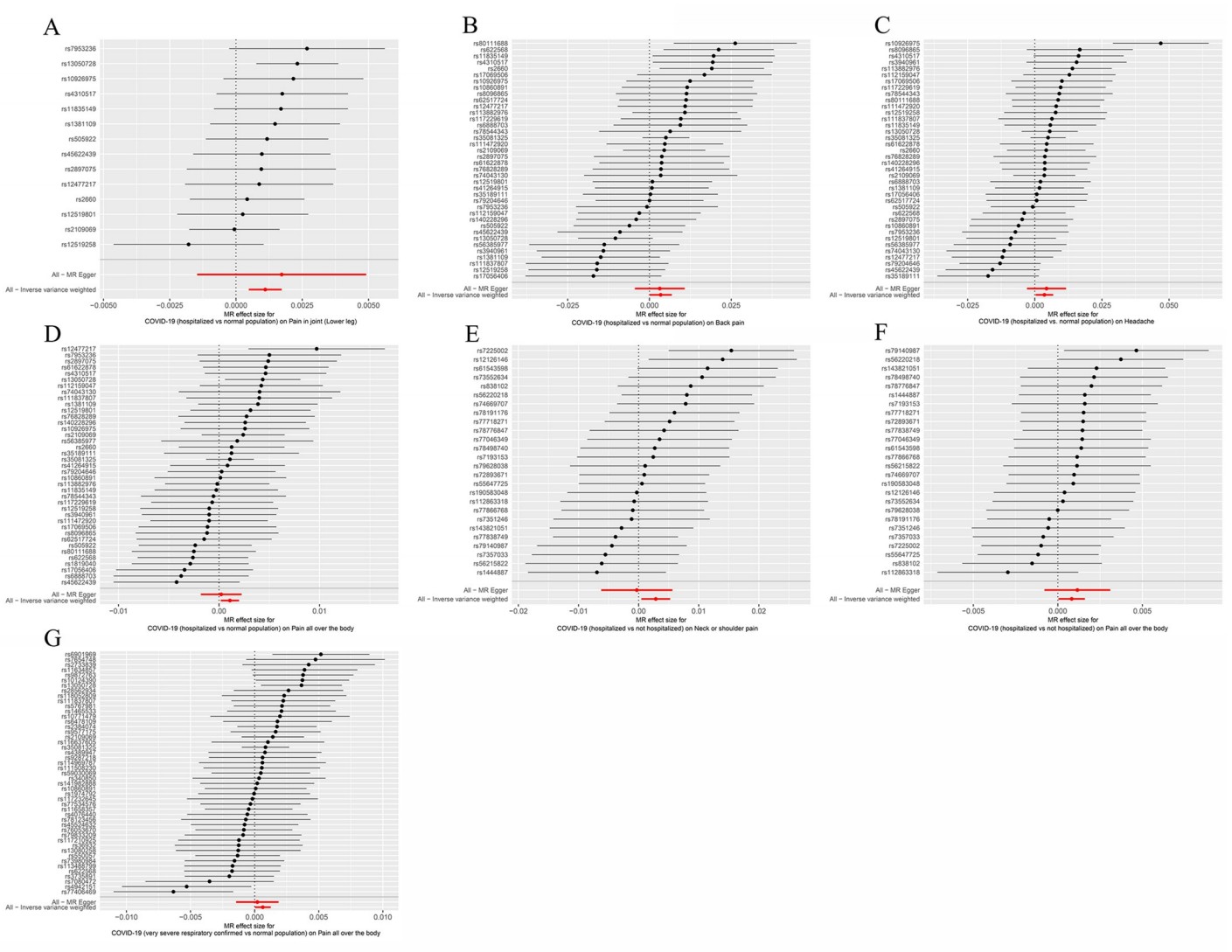

**Fig 4. Forest plot illustrating the MR effect size for different conditions with COVID-19 on pain risk for each of SNPs.** COVID-19 (hospitalized vs. normal population) on **(A)** Pain in joint(Lower leg), **(B)** Back pain, **(C)** Headache, **(D)** Pain all over the body; COVID-19 (hospitalized vs. not hospitalized) on **(E)** Neck or shoulder pain, **(F)** Pain all over the body; **(G)** COVID-19 (very severe respiratory confirmed vs. normal population) on Pain all over the body. MR, Mendelian randomization; SNP, single-nucleotide polymorphism.

experiencing Pain in joint (Lower leg) compared to the normal population. However, for causal effects related to back pain, headache, and pain all over the body, the evidence remains suggestive, necessitating further investigation for confirmation (Table 2, Fig 2B).

Additionally, COVID-19 patients who required hospitalization were found to be at a higher risk of experiencing Neck or shoulder pain (OR: 1.002855; 95% CI: 1.000450–1.005266; IVW P value = 0.019966) and Pain all over the body (OR: 1.000819; 95% CI: 1.000037–1.001601; IVW P value = 0.040095) compared to those who did not require hospitalization (Figs 2C, 3E, 3F, 4E and 4F, S3 Table). Furthermore, patients with severe respiratory confirmed COVID-19 had an increased risk for pain all over the body as compared to the general population, with an OR of 1.000638 (95% CI: 1.000016–1.001260; IVW P value = 0.044438) (Figs 2D, 3G and 4G, S4 Table). Consistent estimates were also obtained from the MR Egger and weighted median

**Table 2. MR estimates of associations between COVID-19 (hospitalized vs. normal population) and bodily pain in various regions across different methods.**

| Trait | IVW | | | MR Egger | | | Weighted median | | | Heterogeneity test | | | | Pleiotropy test | |
|---|---|---|---|---|---|---|---|---|---|---|---|---|---|---|---|
| | β | se | p | β | se | p | β | se | p | IVW Q | p | MR-Egger Q | p | MR-Egger p | PRESSO p |
| Pain in joint | 4.92E-04 | 3.14E-04 | 1.18E-01 | -1.01E-03 | 1.57E-03 | 5.35E-01 | 5.08E-04 | 4.46E-04 | 2.55E-01 | 1.04E+01 | 4.94E-01 | 9.45E+00 | 4.90E-01 | 3.53E-01 | 4.42E-01 |
| **Pain in joint (Lower leg)** | **1.10E-03** | **3.16E-04** | **4.87E-04** | **1.72E-03** | **1.63E-03** | **3.11E-01** | **1.06E-03** | **4.44E-04** | **1.66E-02** | **1.13E+01** | **5.83E-01** | **1.12E+01** | **5.13E-01** | **7.05E-01** | **3.85E-01** |
| Low back pain | 7.68E-05 | 3.66E-04 | 8.34E-01 | -9.27E-04 | 1.94E-03 | 6.41E-01 | -3.61E-04 | 5.19E-04 | 4.87E-01 | 1.01E+01 | 8.16E-01 | 9.78E+00 | 7.78E-01 | 6.07E-01 | 7.53E-01 |
| Low back pain (Lumbar region) | -2.26E-04 | 3.97E-04 | 5.69E-01 | -2.75E-03 | 1.92E-03 | 1.82E-01 | -5.83E-04 | 4.63E-04 | 2.08E-01 | 1.79E+01 | 8.44E-02 | 1.51E+01 | 1.27E-01 | 2.09E-01 | 1.09E-01 |
| Pain in limb (Lower leg) | 1.43E-04 | 3.81E-04 | 7.07E-01 | -2.68E-03 | 1.86E-03 | 1.75E-01 | -3.08E-04 | 4.51E-04 | 4.95E-01 | 1.81E+01 | 1.54E-01 | 1.51E+01 | 2.38E-01 | 1.47E-01 | 1.69E-01 |
| **Back pain** | **3.42E-03** | **1.74E-03** | **4.91E-02** | **3.15E-03** | **3.89E-03** | **4.23E-01** | **4.61E-03** | **2.44E-03** | **5.86E-02** | **5.52E+01** | **2.77E-02** | **5.52E+01** | **2.15E-02** | **9.39E-01** | **2.00E-03** |
| Facial pain | 1.44E-04 | 5.08E-04 | 7.76E-01 | 5.25E-04 | 1.31E-03 | 6.92E-01 | 3.77E-04 | 7.37E-04 | 6.09E-01 | 1.81E+01 | 8.37E-01 | 1.80E+01 | 8.02E-01 | 7.55E-01 | 8.06E-01 |
| **Headache** | **3.58E-03** | **1.66E-03** | **3.14E-02** | **4.36E-03** | **3.72E-03** | **2.49E-01** | **4.49E-03** | **2.10E-03** | **3.27E-02** | **6.03E+01** | **9.19E-03** | **6.02E+01** | **6.98E-03** | **8.15E-01** | **6.20E-02** |
| Hip pain | 6.42E-05 | 1.31E-03 | 9.61E-01 | -7.20E-03 | 3.19E-03 | 3.06E-02 | 5.81E-04 | 1.70E-03 | 7.33E-01 | 4.93E+01 | 6.86E-02 | 4.20E+01 | 1.92E-01 | 1.88E-01 | 7.80E-02 |
| Knee pain | 4.77E-03 | 1.60E-03 | 2.88E-03 | -3.42E-03 | 3.24E-03 | 2.98E-01 | 2.14E-03 | 2.09E-03 | 3.06E-01 | 5.30E+01 | 4.25E-02 | 4.33E+01 | 1.87E-01 | 7.40E-02 | 5.30E-02 |
| Neck or shoulder pain | 2.83E-03 | 1.40E-03 | 4.39E-02 | -3.14E-03 | 3.03E-03 | 3.07E-01 | 1.36E-03 | 2.09E-03 | 5.16E-01 | 3.86E+01 | 3.95E-01 | 3.38E+01 | 5.75E-01 | 3.36E-01 | 2.94E-01 |
| Stomach or abdominal pain | 4.80E-04 | 9.83E-04 | 6.25E-01 | -9.06E-04 | 2.18E-03 | 6.81E-01 | -1.06E-03 | 1.43E-03 | 4.56E-01 | 4.28E+01 | 2.37E-01 | 4.22E+01 | 2.22E-01 | 4.81E-01 | 2.49E-01 |
| **Pain all over the body** | **1.08E-03** | **4.63E-04** | **1.90E-02** | **2.20E-04** | **1.03E-03** | **8.33E-01** | **1.05E-03** | **7.60E-04** | **1.68E-01** | **3.50E+01** | **6.11E-01** | **3.41E+01** | **6.06E-01** | **3.56E-01** | **6.33E-01** |

MR, Mendelian randomization; OR, odds ratio; CI, confidence intervals; IVW, inverse variance weighted.

analyses (S3 and S4 Tables). Nevertheless, due to the P value exceeding 0.0125, all these associations remain in the realm of being suggestive, demanding further empirical validation through additional research endeavors.

To assess the robustness of our results, we conducted a series of sensitivity analyses, including Cochran's Q test, MR-Egger intercept test, and MR-PRESSO global test. While heterogeneity was observed in the Q test analysis for COVID-19 (hospitalized vs. normal population) on Headache (Table 2), no heterogeneity was detected in other analyses. Notably, the MR estimates were not invalidated by this heterogeneity as random-effect IVW was used in this study, which balanced the pooled heterogeneity. Furthermore, while there were potential outliers in some of the IVs, such as COVID-19 (hospitalized vs. normal population) on Back pain, Headache, Neck or shoulder pain, and Pain all over the body, and COVID-19 (hospitalized vs. not hospitalized) on Neck or shoulder pain and Pain all over the body, as well as COVID-19 (very severe respiratory confirmed vs. normal population) on Pain all over the body, further MR-Egger intercept test and MR-PRESSO analysis did not identify any significant outliers (Table 2, S2–S4 Tables). Therefore, there was insufficient evidence to suggest the presence of horizontal pleiotropy in the association between COVID-19 conditions and pain. Additionally, we ensured the eligibility of the SNPs included in the analysis, as they did not show any association with pain or related confounders in Phenoscanner and had F-values greater than 10, indicating the absence of weak instruments (S5 Table). Additionally, the leave-one-out analysis

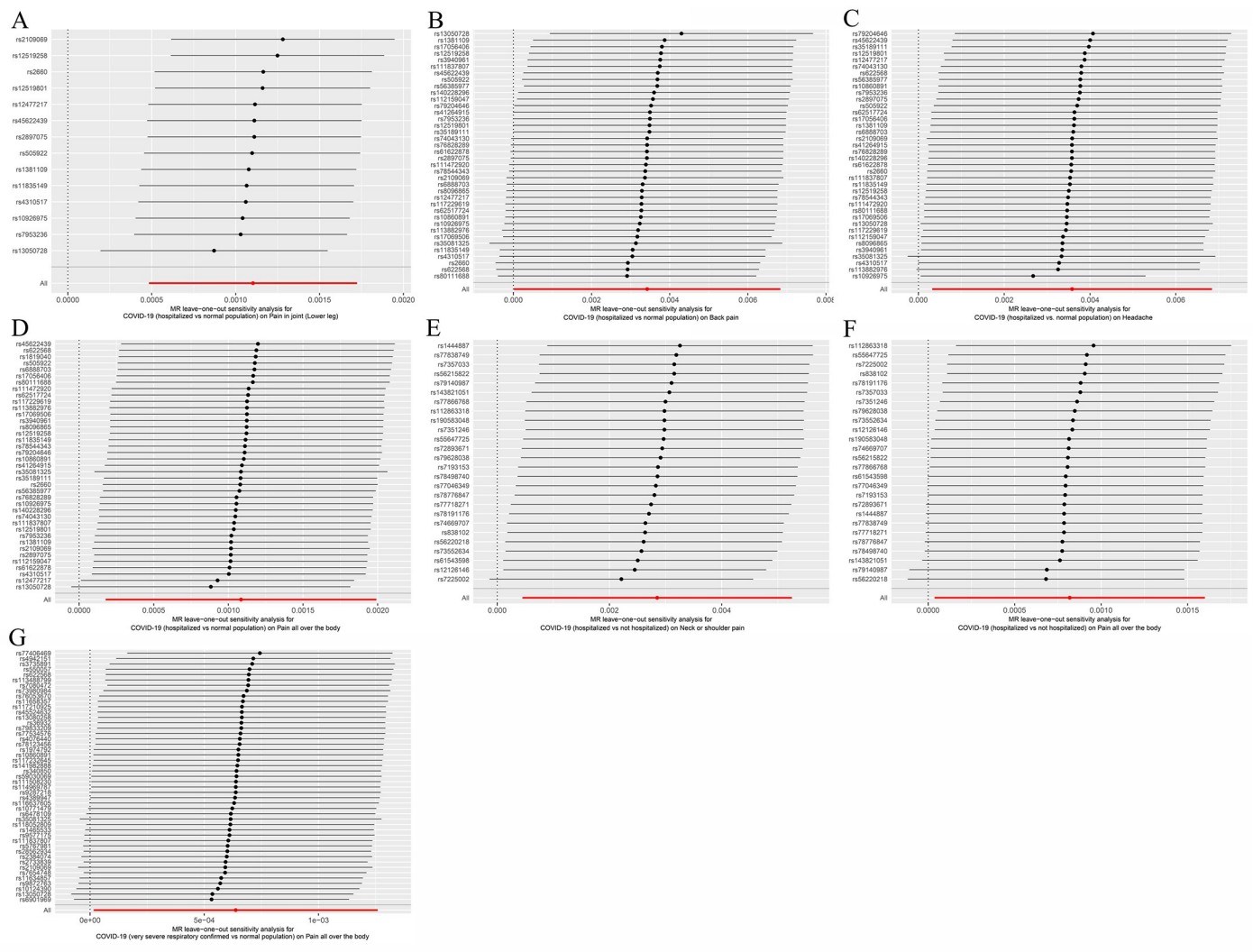

**Fig 5. Leave-one-out sensitivity analysis displaying the MR analysis for different conditions with COVID-19 on pain risk and the effect of excluding individual SNPs.** COVID-19 (hospitalized vs. normal population) on **(A)** Pain in joint(Lower leg), **(B)** Back pain, **(C)** Headache, **(D)** Pain all over the body; COVID-19 (hospitalized vs. not hospitalized) on **(E)** Neck or shoulder pain, **(F)** Pain all over the body; **(G)** COVID-19 (very severe respiratory confirmed vs. normal population) on Pain all over the body. MR, Mendelian randomization; SNP, single-nucleotide polymorphism.

revealed that none of the individual independent IVs exerted an exaggerated influence on the causal effect (Fig 5A–5G). These sensitivity analyses provide further support for the robustness and validity of our findings.

## Discussion

Our study investigated the potential causal relationship between b different COVID-19 prevalence conditions and pain in various regions of the body. A two-sample MR analysis was used to investigate this relationship. IVW methods were used as the primary method to assess causal effects, revealing that hospitalised COVID-19 patients were at higher risk of developing Pain in joint (Lower leg) compared to the normal population. However, due to the stringent Bonferroni correction for multiple testing, the causal effects indicating an elevated risk of

experiencing back pain, headache, and pain all over the body among hospitalized COVID-19 patients when compared to the general population were suggestive in nature. In addition, patients with COVID-19 who required hospitalisation were at a suggestive higher risk of developing neck or shoulder pain and pain all over the body compared to those who did not require hospitalisation. Patients with a severe respiratory confirmation of COVID-19 also had a suggestive increased risk of pain all over the body compared to the general population. Consistent direction was obtained from MR Egger and weighted median analyses, providing further support for the robustness and validity of the study results. To ensure the validity of the results, a series of sensitivity analyses were conducted. While heterogeneity was found in the Q-test analysis of the COVID-19 regarding headache, no heterogeneity was found in the other analyses. The MR estimates were not invalidated by this heterogeneity as the random effects IVW model was used in this study. In addition, the MR-Egger intercept test and MR-PRESSO analysis did not reveal any significant outliers, indicating the absence of pleiotropy.

Our findings suggest that the risk of pain in the overall COVID-19 population is not higher than that in the general population. This may be due to the prevalence of asymptomatic or minimally symptomatic cases of SARS-CoV-2 [30], which could dilute the overall effect of COVID-19. However, our results indicate a possible association between pain and COVID-19 in specific prevalence groups. For instance, hospitalised COVID-19 patients are at a higher risk of developing various types of pain, such as back pain and headache. According to Istvan et al [31]., the frequency of web search terms associated with these pain increased after the COVID-19 outbreak, which is consistent with our findings.

Headache is a common neurological symptom of long-COVID syndrome [32]. Cross-sectional studies reveal that long-COVID headaches are mainly bilateral, frontal or periocular, and compressive [33, 34]. Tension-like headache and migraine-like headache are the most common phenotypes, and refractory headache in the acute phase of infection is a risk factor associated with long-COVID headache [35]. Our results indicate that hospitalised COVID-19 patients are at a suggestive higher risk of developing headaches compared to the general population, suggesting that the risk of prolonged pain may increase with the severity of COVID-19. These findings are supported by evidence suggesting that acute infections can trigger chronic pain, and the severity and duration of the infection must be sufficient to disrupt normal activity [36, 37].

In addition, our results show that hospitalized COVID-19 patients may exhibit a heightened risk of experiencing pain all over the body compared to the general population. Similarly, a comparison between hospitalized and non-hospitalized COVID-19 patients also revealed an elevated risk of pain all over the body in the former group. Furthermore, individuals afflicted with very severe respiratory confirmed COVID-19 may also display an increased likelihood of developing pain all over the body when compared to the general population. This suggests that the presence of body pain may be a prevalent symptom of long-COVID manifestations in severely ill COVID-19 patients. Notably, hospitalized COVID-19 patients appear to be at an elevated risk of experiencing joint pain, particularly in the lower leg, compared to the general population. Additional exposure outcome datasets suggest that knee pain may also be increased, indicating that the knee may be a significant site of attack for long-COVID pain. Several longitudinal studies in Turkey, France, and Italy followed COVID-19 patients for six months after hospital discharge, and approximately 60% of patients continued to experience at least one associated symptom, most commonly fatigue, myalgia, and arthralgia. Arthralgia and myalgia were widespread throughout the body in the majority of patients [38]. A study by Sansin et al. found that myalgia, arthralgia, and back pain were prevalent in patients with COVID, with the knee being one of the primary sites of arthralgia [39]. Furthermore, there was a significant correlation between arthralgia and disease severity. There have been reports of early-

onset arthritis in a few patients after severe acute infection, with this complication more frequently occurring in men and more likely to manifest in the lower leg [40, 41].

Chronic pain caused by COVID-19 can have a significant impact on a person's quality of life, making it imperative to investigate this phenomenon. Such investigations may lead to a better understanding of the pathophysiology of COVID-19, which, in turn, could inform the development of new treatments and interventions. Furthermore, an examination of chronic pain may aid healthcare professionals in identifying patients who are at a higher risk of developing this condition. In our study, we found that hospitalized and severely respiratory confirmed COVID-19 patients were among those at high risk. Early identification of those at risk could prompt timely intervention and management, potentially preventing chronic pain from developing [42]. One potential mechanism through which COVID-19 may contribute to chronic pain is the development of long-COVID. long-COVID is a condition in which individuals experience symptoms of COVID-19 long after they have recovered from the acute phase of the disease [43]. Some individuals with long-COVID report persistent pain or discomfort months after their initial diagnosis of COVID-19. The pain they report is typically diffuse and can affect multiple areas of the body, including the back, neck, and joints [44], which is consistent with our results. Although the relationship between COVID-19 and chronic pain is not yet fully understood, there is evidence to suggest that the mechanisms involved may be multifactorial, involving changes in the immune system, central nervous system, and psychological factors.

Activation of the immune system is a potential mechanism underlying the development of chronic pain. Inflammation is a natural response of the immune system to injury or infection; however, excessive or chronic inflammation can result in tissue damage and organ dysfunction. Inflammation is a hallmark of COVID-19 and plays a pivotal role in the disease's pathogenesis [14, 30]. The cytokine storm, a state of excessive immune response that results in widespread inflammation throughout the body, including muscles, joints, nerves, and the brain, is a characteristic feature of COVID-19 [45]. In COVID-19, the immune system overreacts, leading to the release of large quantities of pro-inflammatory cytokines such as interleukin-6, tumour necrosis factor-alpha, and interferon-gamma [14]. These cytokines can activate other immune cells and recruit them to the infection site, resulting in further inflammation. The sensitization of nerve fibres in the affected region by the release of chemicals from immune cells can lead to an increased response to stimuli, resulting in pain. Inflammation has the potential to activate glial cells within the central nervous system, resulting in increased sensitivity of nerves and intensified pain [46]. Futhermore, multiple studies have indicated that the SARS-CoV-2 virus is capable of infiltrating the central nervous system, potentially resulting in pain [47]. The entry of SARS-CoV-2 into human cells is facilitated by the binding of the spike protein to the angiotensin-converting enzyme 2 (ACE2) receptor, which is distributed across various tissues and organs. This interaction is a crucial pathophysiological mechanism underlying the development of COVID-19 [48]. The binding of SARS-CoV-2 to ACE2 and the consequent disruption of the renin-angiotensin system have been demonstrated to be implicated in endothelial and adipose dysfunction, as well as the initiation of neuronal sensitisation. Such mechanisms may lead to joint pain and osteoarthritis-like changes in bones [38]. Meanwhile, SARS-CoV-2 may also conceivably infect ACE2-expressing cells in the human spinal cord dorsal horn, leading to pain [49, 50]. The development of chronic pain in long-COVID may also involve psychological factors. The stress and anxiety brought about by the pandemic, coupled with social isolation during lockdowns, can trigger a physiological response that worsens pain [15, 16].

In this study, we employed the MR approach to examine the potential association between chronic pain and COVID-19. This method offers several advantages over retrospective and

randomised controlled trials [51]. MR studies utilise genetic markers as IVs, which are randomly allocated at the time of conception and remain unaffected by lifestyle or environmental factors [21]. Therefore, the MR approach can significantly reduce the bias that is typically present in retrospective and randomised controlled trials [20]. Moreover, the MR method can utilise large-scale genetic data to improve statistical power, while the sample sizes required for retrospective and randomised controlled trials can often be prohibitively large [24]. Additionally, MR studies are often more cost-effective than randomised controlled or retrospective studies, as genetic data can be obtained from pre-existing databases or biobanks, and the data collection process is less expensive than that of conducting a randomised controlled trial [24, 52].

Our study has several limitations that warrant consideration. Firstly, we conducted a two-sample MR analysis, assuming the genetic tools used to represent COVID-19 exposure were valid and not influenced by polymorphism. While we performed sensitivity analyses to assess the validity of the results, unmeasured confounders could have influenced the observed associations [24, 51]. Secondly, the study's focus on COVID-19 patients from Western populations limits the generalizability of the findings to other populations with distinct genetic and environmental factors. Thirdly, we did not examine the duration of COVID-19 symptoms and its potential impact on the risk of developing long-term pain. Future studies should investigate the temporal relationship between COVID-19 exposure and pain outcomes. Moreover, while meticulous corrections for multiple testing have been diligently applied, our investigation has illuminated noteworthy associations between specific COVID-19 conditions and chronic pain. Nevertheless, it is essential to acknowledge that the effect sizes derived from the MR analysis are of a relatively modest magnitude. Consequently, we advise circumspection when interpreting these particular conclusions, emphasizing the imperative need for subsequent comprehensive research endeavors and inquiries to corroborate and substantiate these findings. Finally, the availability of data for certain pain outcomes restricted the scope of our analysis. Future studies must investigate additional pain sites and different levels of pain intensity. Confirming the associations observed in this study will require larger epidemiological investigations.

## Conclusion

In conclusion, the detrimental impact of chronic pain on an individual's quality of life underscores the importance of studying its prevalence in the context of COVID-19. Our research sheds light on the potential association between varying degrees of COVID-19 severity and the manifestation of pain across different regions of the body. These findings hold significant implications for clinical management of COVID-19 patients. By identifying those at high risk of developing chronic pain, targeted interventions can be developed to mitigate the onset and progression of this debilitating condition, thereby reducing the burden on individuals, healthcare systems, and society at large.

## Supporting information

**S1 Table. Detailed information pertaining to SNPs in the two-sample MR analysis.**
(DOCX)

**S2 Table. MR estimates of associations between COVID-19 (infection vs. normal population) and bodily pain in various regions across different methods.**
(DOCX)

**S3 Table. MR estimates of associations between COVID-19 (hospitalized vs. not hospitalized) and bodily pain in various regions across different methods.**
(DOCX)

**S4 Table. MR estimates of associations between COVID-19 (very severe respiratory confirmed vs. normal population) and bodily pain in various regions across different methods.**
(DOCX)

**S5 Table. Details of SNPs extracted from exposure and F of them.**
(DOCX)

## Author Contributions

**Conceptualization:** Yuchao Fan.

**Data curation:** Yuchao Fan.

**Formal analysis:** Yuchao Fan.

**Writing – original draft:** Yuchao Fan, Xiao Liang.

**Writing – review & editing:** Yuchao Fan.

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
