## [Decision Letter · Decision Letter 0]

7 Sep 2023

PONE-D-23-21562Causal Relationship between COVID-19 and Chronic Pain: A Mendelian Randomization StudyPLOS ONE

Dear Dr. Fan,

Thank you for submitting your manuscript to PLOS ONE. After careful consideration, we feel that it has merit but does not fully meet PLOS ONE’s publication criteria as it currently stands. Therefore, we invite you to submit a revised version of the manuscript that addresses the points raised during the review process.

We look forward to receiving your revised manuscript.

Kind regards,

Chunyu Liu, PhD

Academic Editor

PLOS ONE

Journal Requirements:

Reviewers' comments:

Reviewer's Responses to Questions

**Comments to the Author**

1. Is the manuscript technically sound, and do the data support the conclusions?

Reviewer #1: Partly

2. Has the statistical analysis been performed appropriately and rigorously? 

Reviewer #1: Yes

3. Have the authors made all data underlying the findings in their manuscript fully available?

Reviewer #1: Yes

4. Is the manuscript presented in an intelligible fashion and written in standard English?

Reviewer #1: Yes

5. Review Comments to the Author

Reviewer #1: This study focuses on establishing a causal relationship between COVID-19 and chronic pain. The authors conducted a two-sample Mendelian Randomization analysis to investigate this causality. Their findings suggest that COVID-19 patients requiring hospitalization face an elevated risk of developing various types of pain compared to the general population. The manuscript is well- written. However, a few points warrant attention:

1) In Table 1, "Sample size" was mistakenly spelled as "Simple size."

2) The authors excluded certain SNPs based on pre-established exclusion criteria. However, the exclusion process lacks elaboration. Please provide more details. For example, how many SNPs were removed due to MAF? What was the final count of SNPs for analysis? Additionally, kindly provide details about the LD reference panel employed for instrumental variable selection.

3) The authors opted for a significance level of 0.05. Given the assessment of numerous statistical associations across diverse exposure-outcome pairs, multipart testing is a matter that needs to be considered.

4) Some significant IVW results did not align with robust MR methods. It would be valuable to check the power of MR analyses.

7) The statement regarding the F-statistic lacks clarity, with no mention of EAF or N in the formula employed. Please check and rewrite the statement for F-statistic.

5) Notably, effect sizes in MR analyses were pretty small, even though some associations proved significant prior to multiple testing correction. Consequently, there is a potential for over-interpretation of IVW method results, potentially leading to a misleading conclusion. It is crucial for the authors to exercise caution in drawing their conclusions.

6. PLOS authors have the option to publish the peer review history of their article (what does this mean?). If published, this will include your full peer review and any attached files.

Reviewer #1: **Yes: **Yuankai Zhang

---

## [Author Response · Author response to Decision Letter 0]

17 Sep 2023

Dear Reviewer,

We would like to express our sincere appreciation for your insightful comments on our manuscript titled "Causal Relationship between COVID-19 and Chronic Pain: A Mendelian Randomization Study". Your constructive feedback is highly valued and we have made every effort to incorporate your suggestions into our work.

After careful consideration of your comments, we have made significant revisions to our manuscript, which are highlighted in red. We have taken every effort to address your concerns and refine our work accordingly. Enclosed please find the revised version, which we respectfully submit for your kind and thorough review.

Thank you once again for your time and effort in reviewing our manuscript. We look forward to hearing from you soon.

Reviewer #1: This study focuses on establishing a causal relationship between COVID-19 and chronic pain. The authors conducted a two-sample Mendelian Randomization analysis to investigate this causality. Their findings suggest that COVID-19 patients requiring hospitalization face an elevated risk of developing various types of pain compared to the general population. The manuscript is well- written. However, a few points warrant attention:

The Reviewers' comment: 1) In Table 1, "Sample size" was mistakenly spelled as "Simple size."

The Author's answer:Thank you for pointing this out. We agree with your suggestion and have modified the word.

The Reviewers' comment: 2) The authors excluded certain SNPs based on pre-established exclusion criteria. However, the exclusion process lacks elaboration. Please provide more details. For example, how many SNPs were removed due to MAF? What was the final count of SNPs for analysis? Additionally, kindly provide details about the LD reference panel employed for instrumental variable selection.

The Author's answer: We sincerely appreciate your astute observation. We wholeheartedly concur with your valuable suggestion and have made the necessary revisions in the appropriate sections as per your guidance.

A comprehensive elucidation of the exclusion procedures pertaining to Single Nucleotide Polymorphisms (SNPs) is outlined below for your reference：

Line 170-177: In adherence to our stringent criteria for SNP selection, a defined set of SNPs has been meticulously curated to function as instrumental variables in the context of our MR study. Detailed information pertaining to these chosen SNPs, which includes Number of SNPs extracted from exposure, Number of SNPs extracted from outcome, Number of SNPs after harmonising with outcome, SNPs removed due to harmonization, and Filtering for SNPs significantly associated with exposure in outcome data , has been meticulously documented in Supplemental Table 1 for comprehensive reference. No SNPs were removed due to MAF.

In addiction, LD reference panel employed for instrumental variable selection was “linkage disequilibrium (LD) r2 of less than 0.001 within a 10,000 kb distance were selected as IVs.” which was list in line: 128-129.

The Reviewers' comment: 3) The authors opted for a significance level of 0.05. Given the assessment of numerous statistical associations across diverse exposure-outcome pairs, multipart testing is a matter that needs to be considered.

The Author's answer: We extend our sincere appreciation for highlighting this aspect. We concur with your valuable suggestion and have duly implemented the necessary revisions in the relevant sections: 

Abstract：line 29: The Bonferroni method was employed for the correction of multiple testing.

Line 31-38: Based on the IVW method, hospitalized COVID-19 patients exhibit a higher risk of experiencing lower leg joint pain compared to the normal population. Meanwhile, the associations between COVID-19 hospitalization and back pain, headache, and pain all over the body were suggestive. Additionally, COVID-19 patients requiring hospitalization were found to have a suggestive higher risk of experiencing neck or shoulder pain and pain all over the body compared to those who did not require hospitalization. Patients with severe respiratory-confirmed COVID-19 showed a suggestive increased risk of experiencing pain all over the body compared to the normal population.

Material and Methods：line 150-155: To mitigate the challenge associated with conducting numerous statistical tests, we implemented a stringent correction method based on the Bonferroni principle, setting the significance threshold at p < 0.0125 (0.05 divided by 4). P-values falling within the interval of 0.0125 to 0.05 were deemed indicative of preliminary indications of potential causal relationships, warranting subsequent validation and confirmation. 

Result: line 193-198: Furthermore, considering the Bonferroni-corrected threshold set at 0.0125 for multiple testing, it is noteworthy that patients hospitalized with COVID-19 exhibit a significantly increased risk of experiencing Pain in joint (Lower leg) compared to the normal population. However, for causal effects related to back pain, headache, and pain all over the body, the evidence remains suggestive, necessitating further investigation for confirmation (Table 2, Figure 2B).

Line 207-209: Nevertheless, due to the P value exceeding 0.0125, all these associations remain in the realm of being suggestive, demanding further empirical validation through additional research endeavors.

Discussion: line 233-240: However, due to the stringent Bonferroni correction for multiple testing, the causal effects indicating an elevated risk of experiencing back pain, headache, and pain all over the body among hospitalized COVID-19 patients when compared to the general population were suggestive in nature. In addition, patients with COVID-19 who required hospitalisation were at a suggestive higher risk of developing neck or shoulder pain and pain all over the body compared to those who did not require hospitalisation. Patients with a severe respiratory confirmation of COVID-19 also had a suggestive increased risk of pain all over the body compared to the general population.

Line 374-379: Moreover, while meticulous corrections for multiple testing have been diligently applied, our investigation has illuminated noteworthy associations between specific COVID-19 conditions and chronic pain. 

Additionally, we have made revisions to Figure 2 and correspondingly adjusted the references to this figure within the text to enhance clarity and accuracy.

The Reviewers' comment: 4) Some significant IVW results did not align with robust MR methods. It would be valuable to check the power of MR analyses.

The Author's answer: We sincerely appreciate your insightful observation. We regret any oversights in those areas and readily acknowledge the errors therein. Subsequently, we have undertaken the necessary revisions in the respective sections:

Material and Methods：line 146-149: Significant substantive outcomes necessitated concurrence among the results derived from MR-Egger, the weighted median, and IVW methodologies in terms of directional implications. In the absence of such alignment, the observed significance would remain predominantly nominal in nature.

Results:line 190-193: However, owing to the incongruity in the directional outcomes between IVW and MR-Egger analyses, the increased risk of knee pain and neck or shoulder pain among patients hospitalized with COVID-19 compared to the normal population was merely nominal (Table 2).

Furthermore, we have made refinements to Figures 2 through 5, along with corresponding amendments in the text regarding the references to these figures and figure legends.

The Reviewers' comment:5) The statement regarding the F-statistic lacks clarity, with no mention of EAF or N in the formula employed. Please check and rewrite the statement for F-statistic.

The Author's answer:We greatly appreciate your valuable feedback. In response to your suggestions, we have revised the statement concerning the F-statistic as follows: line 135-140:

The determination of the attributable fraction of variance ascribed to individual SNPs was computed through the formula: R2 = 2 × β2 × EAF × (1 − EAF)/(2 × β2 × EAF × (1 − EAF) + 2 × SE2 × N × EAF × (1 − EAF)). Simultaneously, the F-statistic was ascertained using the following equation: F = (N − k − 1)/k ×R2 /(1 − R2), wherein 'N' denotes the count of samples subjected to the GWAS, 'k' signifies the number of IVs, and 'R2' characterizes the degree to which IVs expound upon the exposure under investigation.

The Reviewers' comment:6) Notably, effect sizes in MR analyses were pretty small, even though some associations proved significant prior to multiple testing correction. Consequently, there is a potential for over-interpretation of IVW method results, potentially leading to a misleading conclusion. It is crucial for the authors to exercise caution in drawing their conclusions.

The Author's answer: We express our sincere gratitude for bringing this matter to our attention. We wholeheartedly concur with your perspective and, in accordance with your insightful suggestion, have augmented the section discussing the limitations of our study as follows: line 347-353：

Moreover, while meticulous corrections for multiple testing have been diligently applied, our investigation has illuminated noteworthy associations between specific COVID-19 conditions and chronic pain. Nevertheless, it is essential to acknowledge that the effect sizes derived from the MR analysis are of a relatively modest magnitude. Consequently, we advise circumspection when interpreting these particular conclusions, emphasizing the imperative need for subsequent comprehensive research endeavors and inquiries to corroborate and substantiate these findings.

---

## [Decision Letter · Decision Letter 1]

4 Dec 2023

Causal Relationship between COVID-19 and Chronic Pain: A Mendelian Randomization Study

PONE-D-23-21562R1

Dear Dr. Fan,

We’re pleased to inform you that your manuscript has been judged scientifically suitable for publication and will be formally accepted for publication once it meets all outstanding technical requirements.

Kind regards,

Chunyu Liu, PhD

Academic Editor

PLOS ONE

Additional Editor Comments (optional):

Reviewers' comments:

Reviewer's Responses to Questions

**Comments to the Author**

1. If the authors have adequately addressed your comments raised in a previous round of review and you feel that this manuscript is now acceptable for publication, you may indicate that here to bypass the “Comments to the Author” section, enter your conflict of interest statement in the “Confidential to Editor” section, and submit your "Accept" recommendation.

Reviewer #1: All comments have been addressed

Reviewer #2: All comments have been addressed

2. Is the manuscript technically sound, and do the data support the conclusions?

Reviewer #1: Yes

Reviewer #2: Yes

3. Has the statistical analysis been performed appropriately and rigorously? 

Reviewer #1: Yes

Reviewer #2: Yes

4. Have the authors made all data underlying the findings in their manuscript fully available?

Reviewer #1: Yes

Reviewer #2: Yes

5. Is the manuscript presented in an intelligible fashion and written in standard English?

Reviewer #1: Yes

Reviewer #2: Yes

6. Review Comments to the Author

Reviewer #1: (No Response)

Reviewer #2: (No Response)

7. PLOS authors have the option to publish the peer review history of their article (what does this mean?). If published, this will include your full peer review and any attached files.

Reviewer #1: **Yes: **Yuankai Zhang

Reviewer #2: No

---

## [Editor Report · Acceptance letter]

10 Jan 2024

PONE-D-23-21562R1 

PLOS ONE

Dear Dr. Fan, 

I'm pleased to inform you that your manuscript has been deemed suitable for publication in PLOS ONE. Congratulations! Your manuscript is now being handed over to our production team.

Kind regards, 

on behalf of

Dr. Chunyu Liu 

Academic Editor

PLOS ONE